# Drone Optimization in Factory: Exploring the Minimal Level Vehicle Routing Problem for Efficient Material Distribution

Ivan Derpich [1,*] and Carlos Rey [2,*]

1    Industrial Department, University of Santiago of Chile, Santiago 9170124, Chile
2    Departamento de Ingeniería Industrial, Universidad del Bio-Bio, Concepción 3780000, Chile
*    Correspondence: ivan.derpich@usach.cl (I.D.); crey@ubiobio.cl (C.R.); Tel.: +56-989190614 (I.D.)

**Abstract:** The efficient movement of raw materials within organizations is fundamental to maintaining the seamless progression of production processes. However, these logistical operations can inadvertently compromise overall company efficiency, primarily due to the substantial time invested in transporting materials. This paper introduces an innovative mathematical model specifically designed to optimize the transport of raw materials via drones across multiple workstations. This model employs a novel modification of the traditional multi-level Vehicle Routing Problem by incorporating an additional index and accounting for the drone's energy consumption. We employ a widely-recognized solver for practical resolution and compare it with a heuristic algorithm. The resultant strategies offer promising prospects for the organization studied, introducing robust solutions for elevating the efficiency of raw material transportation.

**Keywords:** mathematical model; heuristic; minimum levels; drones in factory

## 1. Introduction

Integrating drones into diverse industrial production processes has become a crucial asset. Many companies are harnessing this technology to enhance efficiency and minimize expenses. By enabling the execution of tasks with increased speed, precision, and safety, drones significantly boost productivity while reducing the likelihood of human errors. Moreover, they can access challenging or hazardous locations, mitigating the risks associated with workplace accidents. Consequently, it is imperative to examine the operational aspects of drones, encompassing both their movement dynamics and energy consumption patterns.

Drones exhibit exceptional versatility in various industrial sectors due to their ability to maneuver both vertically and horizontally. Vertical movement enables drones to reach significant heights, proving particularly advantageous in photography fields [1,2]. In this context, drones can effectively inspect buildings and structures with accessibility challenges. Similarly, their capacity for horizontal movement renders them a highly efficient resource for transportation and logistics tasks [3]. Drones can seamlessly transport materials and products across multiple points within a given space. This highlights the inherent flexibility of drones and their substantial contributions to diverse production processes.

Currently, using drones to transport raw materials inside buildings has become an increasingly common practice in the industry. Drones allow both horizontal and vertical routes to transport or collect essential items for the company. This technology means that company personnel do not have to travel to perform these tasks, which increases efficiency and makes it possible to make better use of time for other operational activities.

Optimizing routes for transportation and logistics tasks is crucial, as it ensures efficiency and curtails costs [4]. Through strategic route planning, extra trips can be eliminated, travel distances shortened, and cargo capacities maximized for each flight, thereby maintaining overall process efficiency. Furthermore, minimizing routing reduces waiting and delivery times, fostering seamless integration within a company's productive workflow.

Drones have the potential to transform manufacturing operations in the future radically. Their ability to automate inspection and maintenance tasks could significantly reduce downtime and improve efficiency. Drones equipped with advanced scanning and imaging technology could perform accurate and fast inspections of machinery and equipment, detecting problems that may not be visible to the naked eye and thus preventing costly breakdowns and stoppages. Moreover, with the growing evolution of robotic and drone technology, in the future, drones can obtain a more active role in the production line, handling and moving small parts or even participating in assembly processes. The adoption of drones can also improve job safety by reducing the need for workers to perform dangerous or high-risk tasks. These advantages could lead to significant cost and time savings while improving quality and safety in the manufacturing industry.

In order to implement drone-based technological solutions, it is essential to account for and manage variables related to the transported objects and the associated operational costs. Energy expenses (batteries, fuel, others) must also be considered, as they are directly linked to the load size and travel distances. The objective is to minimize operational costs while ensuring the transported items arrive in optimal condition. A route planning design that considers these constraints is required. This can be accomplished through adjustments and Vehicle Routing Problem (VRP) formulations.

This paper proposes employing drones to transport raw materials between various workstations within a company. To tackle this issue, we have devised mathematical models inspired by the well-established Vehicle Routing Problem. Additionally, we introduce a heuristic approach and compare it with a classic solver. Our contribution is a novel variant called Minimal Levels VRP, which incorporates an extra index based on the factory's floor count, aiming to minimize drone trips between floors and subsequently reduce vertical distances. For multiple reasons, including potential accidents, Unmanned Aerial Vehicles (UAVs) should fly from the base to a designated floor and then return to the base. Simultaneously, pertinent aspects, such as the calibration of the penalty factor, are integrated into the model.

The experiment is executed on a large scale, utilizing a fleet that simultaneously transports materials to the 128 workstations, also referred to as nodes, which constitute the distribution network. The model is then solved to determine optimal drone routes within the factory. The test model is conducted across various scenarios to assess the effectiveness of available resolution models for optimization problems. This approach enables the extrapolation of results for more complex networks, contributes to the real-world implementation of the model, and evaluates the efficacy of these methods in internal delivery dispatch.

The proposed mathematical model incorporates a new index within its variables, providing a notably improved level of precision and efficiency compared to previous models. This index is crucial for generating optimized routes that innovative assignments can effectively implement on each manufacturing floor under study. In this way, it provides a robust theoretical framework and offers practical solutions that can be easily applied in the actual production plant setup, thus improving operational efficiency and productivity.

The following Section 2 provides a brief review of the existing literature on VRP models, their origins, and applications; in particular, the available literature on VRP models applied inside factories or industries. Section 3 provides a description of the problem and the approached scenario. Section 4 presents the mathematical models used. In Section 5 a heuristic for the problem is defined. In Section 6 the computational experiments are presented. Finally, Section 7 presents the conclusions.

## 2. Literature Review

The earliest application of the VRP focused on gasoline distribution to gas stations. This formulation assigned trucks to fuel stations to fulfill demand while minimizing the trucks' travel distance. Subsequently, Clarke and Wright [5] formally generalized an algorithm capable of serving a geographically distributed group of clients from a depot using

a fleet of trucks, thus creating the Vehicle Routing Problem (VRP). Numerous applications, formulations, and variants of the VRP have been developed to represent real-world scenarios. Notable examples include the Vehicle Routing Problem with Time Windows (VRPTW), which seeks to minimize route distances within a specified time frame [6,7]; the Multiple Depot Vehicle Routing Problem (MDVRP), in which multiple depots are geographically distributed along with associated vehicle fleets [8]; and the Capacitated Vehicle Routing Problem (CVRP), employing vehicles of identical characteristics from a single depot with known demand requirements. These variants apply to various vehicle types, but this article specifically focuses on drones, which are constrained by their limited capacities. This particular challenge has sparked interest in the formulation and resolution algorithms for CVRP as VRP models must be adapted to accommodate the unique characteristics of drones.

The utilization of drones for deliveries has been previously investigated, revealing their potential to decrease transportation costs and reduce delivery times [9]. Moreover, drones contribute to sustainable processes by lowering $CO_2$ emissions [3]. However, various drone models differ significantly due to the specific requirements for transported products and the locations where transportation occurs. These variables can constrain drone capacity, as exemplified by transporting organs or blood products. In [10], the feasibility of using drones for blood transport is explored, taking into account not only the drone's specifications (characteristics, costs, others) but also the implications of drone transport for blood, which must arrive in optimal conditions at its destination. In [11], a network of drones is optimized for transporting external defibrillators, a task with fewer transportation limitations than biological materials since defibrillators are more durable and do not require the same level of care and precautions. In [12], drones serve as a last-mile distribution method for rescue operations during disasters, demonstrating their effectiveness by reducing response times in catastrophic events.

In general, efficient delivery systems that reduce cost and time are crucial for addressing business operations and logistics challenges. Authors such as [13] have suggested that effective distribution management improves service levels in various industries. By better satisfying customer needs, businesses can increase sales, gain a larger market share, and ultimately achieve higher profits.

Although the use of drones in large-scale external networks has been extensively analyzed and associated with numerous applications, the implementation of these technologies within indoor environments still needs to be explored. Ref. [13] investigates a problem where drone routing with limited capacities is applied to distribute goods inside a single-level warehouse. Similarly, ref. [14] presents the development of software systems for using drones in internal transportation, featuring energy-efficient drones and lightweight cargo-handling devices. However, these studies need to consider applications for multi-level companies where different stages of the production process take place on each floor, a common business practice. Additionally, they do not account for the limited budget available for drone acquisition, as the purchase cost for these devices is often high, rendering state-of-the-art technologies not always accessible.

Similar to the VRP, its variant, the CVRP, is an NP-Hard problem, as its complexity grows with the increasing number of nodes to be visited. Finding optimal solutions using exact techniques for large networks in combinatorial optimization problems is challenging, and has led to utilizing two-stage heuristic and metaheuristic methods that find solutions based on node clustering [13]. The Lin–Kernighan heuristic, one of the most efficient methods, has laid the foundation for highly effective heuristics, such as the Lin–Kernighan-Helsgaun method [15]. Another popular heuristic for solving VRP is the Clark and Wright Algorithm, also known as the "Saving Algorithm", which enables the identification of optimal or near-optimal solutions for routing problems. In general, various resolution algorithms differ in the number of steps and their implementation. For instance, this article utilizes a two-phase heuristic to sort and assign routes. However, an alternative method could be achieved by assigning and sorting, resulting in a different approach.

## 2.1. Drones Inside the Factory in the Literature

Table 1 shows studies in which drones are used in processes associated with manufacturing and factory logistics. This search was carried out in the Web of Science (WOS) index and the keyword "drones in factory" were used. The search yielded 45 results and the most outstanding articles are those that were registered in this table. The article developed by [16] is the most prominent as a review of the state of the art, they indicate that there are very few applications for the use of drones within the factory, and many studies that are still promising. In the article developed by [17], drones are used for merchandise recognition through infrared technology located in the drone and a data acquisition and control system is developed for the DHL postal company. The article [18] shows the use of drones in security tasks both inside and outside the factory, while the article [19] shows an application of drones for quality inspection in the factory. Finally, the articles [20–22] show applications of the Internet of Things in the industrial field.

**Table 1.** Summary of applications for the use of drones in factories and logistics.

| Author | Methodology/Method | Description | Application Area |
|---|---|---|---|
| [16] | Simulation virtual, Automatic drones for factory, Virtual simulation, Inspection | The current and potential applications of drones in product manufacturing applications of drones in product manufacturing are explored and the opportunities and challenges involved are examined | Review |
| [17] | Inventory management, Business adoptions, Smart city, Inventory control, | The development of a Supervision, Control and Data Acquisition (SCADA) system is proposed for the inventory management of the DHL company, using drones | Inventory management |
| [18] | Smart city, Factory security, Surveillance, Indoor environment, Minimize the human factor | This article introduces the application of drones for specific tasks in industrial areas, indoors and outdoors. | Factory safety |
| [19] | A virtual simulation of drone lights at a manufacturing plant was developed to assess the potential of drones for thermal inspection of machines | This document informs about how to use automatic drones for factory inspections | Factory quality inspection |
| [20] | Smart logistics, Industrial IoT, Scheduling, Super-resolution, Self-adaptive control, Algorithm for large-scale surveillance | This paper shows an application that uses drones equipped with CCTV cameras for the container supervision | Smart logistic |
| [21] | Trajectory optimization, UAV communication, algorithm search, unmanned aerial vehicle motion planning optimization techniques | Major research challenges are reviewed and reveal implementation gaps in IIoT applications in logistics and manufacturing | Internet of Thinks |
| [22] | Trajectory optimization, UAV communication, algorithm search, unmanned aerial vehicle motion planning optimization techniques | A use case is built and then a modeling environment is described and simulated communications are evaluated | Industrial Internet of Thinks |

## 2.2. Our Contribution

In previous work, a 3D-VRP model exposed in [9] was developed using an objective function that uses the distance between the nodes as a cost parameter of the objective function, which was validated with six drones in a single case of application. In this paper, the VRP MinLE model is presented using an objective function that minimizes the energy consumed by the drone equipment, calibrating a parameter that penalizes the energy consumption generated by vertical movement in relation to the energy consumption consumed by horizontal movement.

This new model, with its one-or-two-floor rule, represents a shift from prior practices, as it introduces a new, more controlled approach to drone operation, focusing on specific

floors and minimizing any potential for overreach. It is a strategy that merges precision with efficiency, ensuring the drone's energy and resources are optimally used. It is worth noting that this updated model not only benefits the operation of drones; it also avoids any collateral damage within the company's production processes, which is highly beneficial in this scenario.

The contribution of this paper aims to extend the studies for UAV routing inside the industries with more than one level, considering the goal that drones fly from the base to a factory workstation and ideally return without making stops on other floors. This is because it is desirable that at any given time there be a minimum of drones on a floor. Two resolution methods available for implementation and the differences that make them viable in practice are evaluated.

### 3. Problem Configuration

This article investigates the feasibility of using drones to transport necessary materials from a factory's warehouse to various workstations, thereby minimizing delivery time. The proposed model is tested in a medium-sized, four-story shoe factory.

The factory comprises 128 workstations, distributed evenly with 32 workstations per floor, grouped into 16 work areas. Each workstation is assigned a list of tasks related to shoe manufacturing that must be completed during the day. The materials required for these tasks are distributed in 2 kg bags. Since the tasks depend on customer demand, material delivery must be consistently scheduled according to the requested batches. Because the floors are the same with the same surface area, 32 jobs were assigned by floor, but they are grouped by processes within the manufacturing process, following product flow. Thus, on the fourth floor the leather is cut, on the third floor upper part of the shoe is sewn, on the second floor the sole and the heel are made of the shoe, while on the first floor, the shoe is assembled by joining the upper and sole and tacos.

Currently, the material distribution process is carried out by one person who collects the bags of material from the warehouse on a cart. Subsequently, a team of eight people, two on each floor, distributes the bags manually. This process takes, in the worst case, two hours to cover all the workstations, a duration considered excessive, especially for the morning shift, which is the most demanding and requires work to commence as quickly as possible. After completing the tasks, the finished products are collected on carts and transported to the warehouse, where they are checked and dispatches are rescheduled. The collection process also takes, in the worst case, two hours.

This excessive time is due to two effects: first, the dealers attend to problems reported by operators at the beginning of the shift such as lack of tools, minor failures of machines, lack of materials, etc.; and second, time is lost in conversations personal. The implementation of the dispatch with drones avoids this exchange and leaves it to specialized personnel from the maintenance and supply area who deal with failures of equipment and lack of materials, respectively. The personnel that are released from this task are assigned to load the drones in the dispatch area. Considering that a worker of this type in Chile represents a company expense of 9.000 euros annually. It can be established that the saving of eight workers due to the use of drones brings a saving of 72.000 euros per year.

Figure 1 illustrates the layout of the first floor, which houses raw materials, products under development, the finished product warehouse, and the request preparation room. The second, third, and fourth floors have identical workstation layouts, as shown in Figure 2. The arrows between work areas represent the transit routes for drones. Workstations or nodes are numbered, making up the distribution network to be fulfilled.

For material transportation, 15 drones are considered, with specifications based on [23,24]. These drones can carry loads of up to 25 kg, and have a weight of 5 kg, leaving 20 kg available for load transportation.

We propose mathematical models and a heuristic approach to tackle the above-mentioned problem. In the following section, we present two mathematical models inspired by VRP and its adaptation to the three-dimensional challenge.

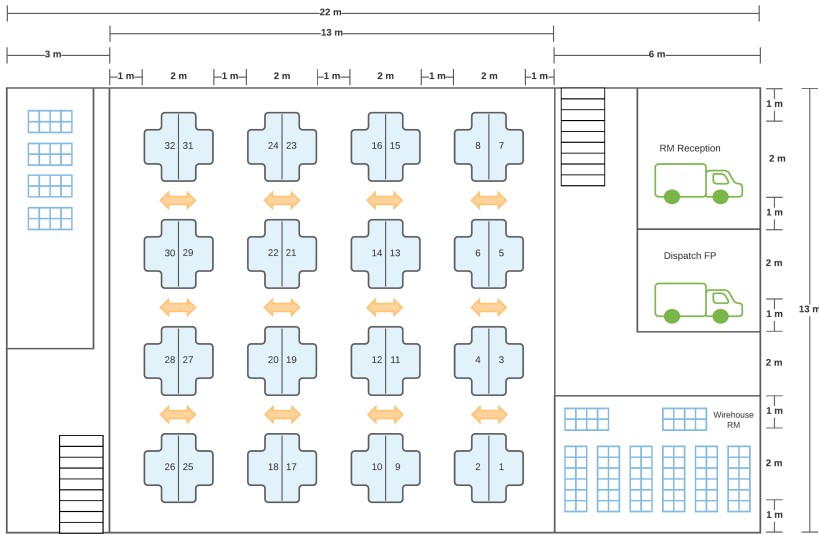

**Figure 1.** Distribution of level 1 in the factory.

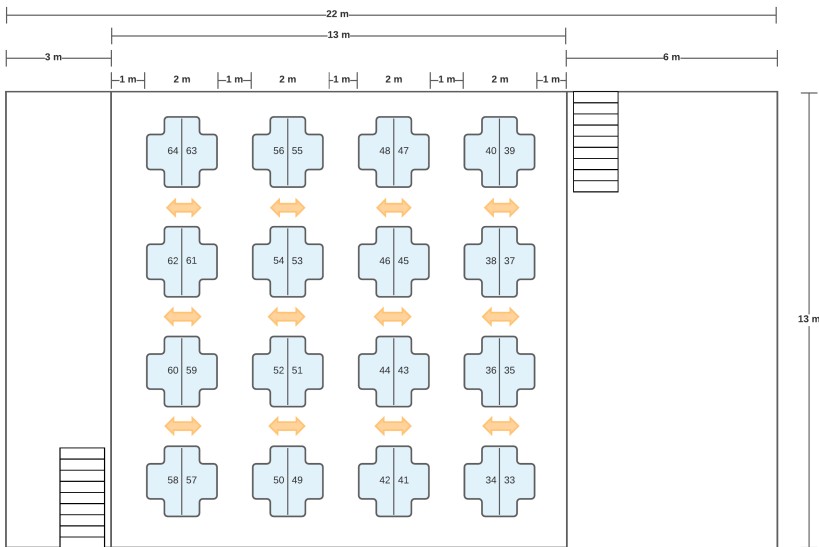

**Figure 2.** Distribution of level 2, 3, and 4 in the factory.

## 4. VRP Models

First, we consider the three-dimensional VRP for drone routing, or three-dimensional CVRP, considering the limited load capacity of drones. Building upon this model, we extend the VRP for minimal levels, contributing to the existing routing models. This study employs this extended model for conducting tests and collecting results. Both models, the 3D VRP and the VRP for minimal levels are characterized as follows.

### 4.1. 3D VRP Model

The 3D VRP model, derived from [25], enables the determination of material distribution routes between the warehouse and workstations while minimizing the associated energy cost. The objective function (OF) minimizes the total distances traveled independently in both the horizontal plane and the vertical axis.

In order to establish mathematical models, it is imperative to define parameters relevant to the routing problem. Consider $G = \{V, A\}$ as a graph where $V = \{0, \ldots, N\}$ represents an assortment of workstations or nodes, and $A$ is a set of arcs $(i, j)$ subject to the condition that $(i, j) \in V$ with $i \neq j$. The workstation with the index 0 is considered the point of origin where all drones must leave. Each arc $(i, j)$ is characterized by two distinct costs: $h_{ij}$, signifying the horizontal distance between workstations $i$ and $j$, and $b_{ij}$,

representing the vertical distance between the same. Every workstation is associated with a demand $d_i$, given that $i \in V$. Finally, a set of drones $Q = \{0, \ldots, M\}$ have a capacity $C$ and a value $p$, designating the maximum number of workstations a drone can service.

In order to describe the mathematical model, a set of variables corresponding to the parameters delineated above needs to be defined. Initially, a binary variable $x_{ij}^k$ is defined to be one if drone $k$ is allocated to arc $(i, j) \in A$, and zero otherwise. Secondly, a variable $y_{ij}$ is defined to be one if the arc is incorporated into the solution and zero otherwise. Finally, a variable $u_i$ is defined to correlate with the position in the path for node $i$. The model, thus proposed, is as follows:

The 3D VRP is detailed as follows.

Objective function

$$MinZ = \sum_{i=0}^{N} \sum_{j=0}^{N} (1 - \alpha) \ h_{ij} \ y_{ij} + \ \alpha \ b_{ij} \ y_{ij}$$

The objective function considers a penalty parameter that constrains vertical movements due to their higher associated risk of accidents and failures in the work journey. Here, we define the parameter $\alpha$, a penalization factor for the movements considered in the model. This parameter between 0 and 1 has been intricately incorporated into the mathematical model to effectively manage these vertical movement-related risks.

Subjected to:

Constraint (1) ensures that only one drone can be assigned per route.

$$\sum_{k=1}^{M} x_{ij}^k = y_{ij} \ \ \forall \ (i, j) \in A \tag{1}$$

Constraint (2) mandates that each drone arriving at a node comes from only one node.

$$\sum_{i=0}^{N} y_{ij} = 1 \ \ \forall \ i \in V \tag{2}$$

Constraint (3) stipulates that each drone must go to a single node.

$$\sum_{j=0}^{N} y_{ij} = 1 \ \ \forall \ i \in V \tag{3}$$

Constraints (4) and (5) guarantee that all drones leaving the warehouse must return to the same location.

$$\sum_{j=0}^{N} y_{0j} = M \tag{4}$$

$$\sum_{i=1}^{N} y_{i0} = M \tag{5}$$

Constraints (6) and (7) ensure that each drone is not used more than once.

$$\sum_{j=1}^{N} x_{0j}^k = 1 \ \ \forall \ k \in Q \tag{6}$$

$$\sum_{i=1}^{N} x_{i0}^k = 1 \ \ \forall \ k \in Q \tag{7}$$

Constraint (8) prevents each drone from exceeding its maximum payload capacity.

$$\sum_{k=1}^{M} d_j \times x_{ij}^k \leq C \quad \forall\, (i,j) \in A \tag{8}$$

Constraint (9) ensures route continuity.

$$\sum_{i=0}^{N} x_{ia}^k = \sum_{j=0}^{N} x_{aj}^k; \quad \forall\, k \in Q; \quad \forall a \in V \tag{9}$$

Constraint (10) is a flow restriction, limiting the drone's route sequence by the pair of nodes that must be visited.

$$u_i - u_j + p \times y_{ij} \leq p - 1 \quad \forall\, (i,j) \in A \tag{10}$$

Finally, constraints (11) and (12) indicate that the problem's variables must be binary.

$$x_{ij}^k \in \{0,1\} \quad \forall\, (i,j) \in A; \forall\, k \in Q \tag{11}$$

$$y_{ij} \in \{0,1\} \quad \forall\, (i,j) \in A \tag{12}$$

### 4.2. VRP Model for Minimal Levels (VRP MinLE)

In order to minimize the visited levels during a drone's flight, it becomes paramount to integrate energy consumption into the objective function. This requires clearly defining the energy costs corresponding to the drone's horizontal and vertical maneuvers. The energy expenditure linked with the drone's vertical movement between nodes $i$ and $j$ is depicted by $VE_{ij}$ in the Equation (13). Conversely, the energy costs related to horizontal movement between the same nodes are denoted by $HE_{ij}$ in Equation (14).

$$VE_{ij} = ECCV \ w_{ij} \ b_{ij} \ t_{ij} \tag{13}$$

$$HE_{ij} = ECCH \ w_{ij} \ h_{ij} \ t_{ij} \tag{14}$$

where $ECCV$, $ECCH$ correspond to the energy consumption cost for vertical and horizontal movement, respectively; $w_{ij}$ represents load in kilograms transported in arch $ij$ by a drone; $b_{ij}$ and $h_{ij}$ represents the vertical and horizontal distance traveled in the nodes $i$ and $j$; and finally, $t_{ij}$ is the drone traveling time between node $i$ and node $j$. Using the generic formula for velocity as a substitution to simplify Equations (13) and (14), we obtain Equations (15) and (16).

$$VE_{ij} = ECCV \ w_{ij} \ \frac{b_{ij}^2}{v_{ij}} \tag{15}$$

$$HE_{ij} = ECCH \ w_{ij} \ \frac{h_{ij}^2}{v_{ij}} \tag{16}$$

Thus, the formula, integrating both parts, vertical and horizontal for energy consumption, which we will call $ET$, is the following:

$$ET = VE_{ij} + HE_{ij} = ECCV \ w_{ij} \ \frac{b_{ij}^2}{v_{ij}} + ECCH \ w_{ij} \ \frac{h_{ij}^2}{v_{ij}} \tag{17}$$

It should be noted that this formula is used in the objective function divided into a vertical part and a horizontal part and the vertical part is penalized by the factor $\alpha$.

Here, $v_{ij}$ is the horizontal velocity of the drone in arc $ij$, whose average value within the factory is $3\,\frac{m}{s}$. In the case of the problem analyzed, the values for the cost of consumption are: $ECCH = 1.076 \times 10^{-5}\,\frac{\$USD}{kg \times m \times s}$ and $ECCV = 1.846 \times 10^{-5}\,\frac{\$USD}{kg \times m \times s}$. It is important to note that energy consumption is not linear; however, this document has been linearized.

Based on the 3D VRP, the VRP model is planned for minimal levels. In this model, minimal traveling distances between levels are incorporated, minimizing the performed trips between these, with the purpose of generating simpler routes.

The new VRP MinLE model considers a four-dimensional variable. Let $O = \{0, \dots, L\}$ be a set of building levels or floors associated with the problem; the variable $x_{ij}^{kl}$ indicates with a value one if the drone $k \in Q$ is assigned to the trip from the arc $(i, j) \in A$ in level $l \in O$, and with a value of zero otherwise. The model is presented in the following equations:

Objective function

$$MinZ = \sum_{i=0}^{N} \sum_{j=0}^{N} ((1 - \alpha) \times ECCH\ w_{ij}\ \frac{h_{ij}^2}{v_{ij}})\ y_{ij} + (\alpha \times ECCV\ w_{ij}\ \frac{b_{ij}^2}{v_{ij}})\ y_{ij}$$

Subjected to:

$$\sum_{k=1}^{M} x_{ij}^{kl} = y_{ij}\ \ \forall (i, j) \in A;\ \ l \in O. \tag{18}$$

$$\sum_{j=1}^{N} x_{0j}^{kl} = 1\ \ \forall k \in Q;\ \ l \in O. \tag{19}$$

$$\sum_{i=1}^{N} x_{i0}^{kl} = 1\ \ \forall k \in Q;\ \ l \in O. \tag{20}$$

$$\sum_{k=1}^{M} d_j x_{ij}^{kl} \leq C\ \ \forall (i, j) \in A;\ \ l \in O. \tag{21}$$

$$\sum_{i=0}^{N} x_{ia}^{kl} = \sum_{j=0}^{N} x_{aj}^{kl}\ \ \forall k \in Q;\ \ l \in O; a \in N \tag{22}$$

$$x_{ij}^{kl} \in \{0, 1\}\ \ \forall (i, j) \in A;\ \ l \in O;\ \ k \in Q \tag{23}$$

In which index $l$ depends on the floor where the destination node is located. The functionality of these restrictions follows the same logic as the previous model. However, it is now incorporated the location of the node within the factory. In addition, the Equation (24) is added for the fleet drones to visit the minimum number of levels.

$$\sum_{i=1}^{N} \sum_{l=1}^{M} x_{ij}^{kl} = 1\ \ \forall k \in Q\ \ j \in N \tag{24}$$

This formulation increases the number of variables proportionally to the floor levels. We would pass effectively from a case with 128 workstations and 245.760 variables to 983.040.

A heuristic is a problem-solving approach that can generate solutions quickly. In the next section, we provide a detailed description of the heuristic process required to obtain solutions for the problem at hand. This heuristic process is essential to efficiently generate feasible solutions within a short amount of time.

## 5. Heuristic Approach

Two fundamental processes are employed to generate a heuristic solution: clustering and route generation. The first process, clustering, involves assigning workstations to

a finite number of drones. The second process, route generation, focuses on creating valid routes for each drone.

Algorithm 1 shows the general processes used to solve the problem. Initially, the algorithm receives the number of drones $m$, the workstations or "nodes", and the maximum capacity of each drone $C$ as input parameters. In line 1 of the algorithm, the empty solution structure is defined. Line 2 employs the k-means clustering algorithm, a widely used parametric algorithm in the literature, effectively generating $m$ clusters based on the three-dimensional distances between nodes. Lastly, lines 4, 5, and 6 generate routes for each cluster established by the K-Means algorithm. The *forEach* statement takes a cluster (referred to as *setNodes*) and iteratively solves the subproblem using a Solver TSP, ultimately adding the path to the solution.

---

**Algorithm 1:** Heuristic Algorithm

**Input:** $m$ , $nodes = \{i = 0, \dots, n\}$, $C$
**Result:** Solution (Routes Set)
1 Solution $\leftarrow \varnothing$;
2 Clusters $\leftarrow$ K-Means(*nodes*, $m$) ;
3 **foreach** *setNodes* $\in$ *Clusters* **do**
4     routeDrone $\leftarrow$ SolverTSP(setNodes, $C$) ;
5     Solution $\leftarrow$ Solution + routeDrone
6 **end**
7 return Solution;

---

The Lin–Kernighan heuristic (LKH) library, a widely-used tool in operations research, is employed to address the traveling salesperson subproblem. This library efficiently implements the LKH for solving the TSP [15]. By executing the library with a subset of workstations and diving the traveling salesman problem with pickups and deliveries (TSPPD) while setting pickups to 0, feasible routes can be obtained for each subproblem.

## 6. Computational Experiments

### 6.1. Experiment Protocol

The mathematical models presented in this article were solved using a conventional solver, specifically CPLEX 12.5.1. This solver was also employed to obtain the penalty factor $\alpha$. The solver was set with a time limit of one hour of execution (3600 s).

The heuristic was implemented in Python 3.0. For the clustering process, the K-Means algorithm from the Sklearn 1.2 library [26] was utilized. In contrast, the LKH version 3 library was employed for route generation, as discussed in the previous section. The LKH library is implemented in the C language, and its executable is invoked iteratively through the Python script to ensure proper communication and accurate solution generation.

The tests of the model were performed in a computer with a processor Intel® Xeon® E5–2660 v2 (8 cores) and 12 GB of RAM, including the CentOS Linux operative system, version 7.5.1804.

To test the effectiveness of the model in different scenarios, different numbers of workstations or nodes were chosen, thus generating networks of different complexity. These scenarios were generated randomly using a uniform distribution to choose the workstations in each plant; the only condition is the equal distribution on four floors. The smaller instances are useful when we consider smaller factories, scenarios where not all workers are present, or when there is a lower demand and not all stations are needed to participate. On the other hand, some instances also limit drone loading, when transporting both raw materials from warehouses, as well as in-process and finished products.

The generated scenarios for simulation are detailed in Table 2, where scenario number 8 corresponds to the original problem of the plant under study. In Appendix A the simulated scenarios are shown.

**Table 2.** Scenarios used for the application of the model.

| Scenario Number | Nodes | Workstations by Floor | Drone | Drone Capacity (Kg) |
| :---: | :---: | :---: | :---: | :---: |
| 1 | 16 | 4 | 5 | 8 |
| 2 | 32 | 8 | 10 | 8 |
| 3 | 48 | 12 | 10 | 10 |
| 4 | 64 | 46 | 10 | 14 |
| 5 | 80 | 20 | 10 | 14 |
| 6 | 96 | 24 | 15 | 20 |
| 7 | 112 | 28 | 15 | 20 |
| 8 | 128 | 32 | 15 | 20 |

### 6.2. Penalization Factor Calibration

Obtaining the appropriate calibration value for the penalty $\alpha$ contributes to reducing the total cost associated with transportation, because the vertical distances traveled, which have a higher cost, are reduced. This parameter represents the ratio of the increase in costs to reco when traveling distances on the vertical axis versus in the horizontal plane. To determine the value of the parameter, the model is run in CPLEX for different values of $\alpha$ within [0.1, 0.9] with steps of 0.1 to facilitate subsequent calculations and interpretation of the result, as well as to avoid extreme values (0.1 or close).

Table 3 shows the values obtained for the displacement distances for different values of $\alpha$, and these are illustrated in Figure 3.

**Table 3.** Total Traveled distances (mt) at different variations of the $\alpha$ parameter.

| $\alpha$ | XY (mt) | Z (mt) | Total (mt) |
| :---: | :---: | :---: | :---: |
| 0.1 | 635.77 | 243 | 878.77 |
| 0.2 | 589.92 | 231 | 820.92 |
| 0.3 | 562.50 | 241 | 803.50 |
| 0.4 | 602.09 | 197 | 799.09 |
| 0.5 | 593.39 | 188 | 781.39 |
| 0.6 | 623.76 | 170 | 793.76 |
| 0.7 | 570.45 | 168 | 738.45 |
| 0.8 | 671.41 | 158 | 829.41 |
| 0.9 | 688.32 | 164 | 852.32 |

The objective is to find the value that minimizes the total distances traveled by the drones. A considerable decrease in the total distances traveled is observed for $\alpha = 0.7$, where the lowest value is reached, corresponding to 738.45 m. With a mean of $\mu = 810.8455$ m, this value for the parameter has a negative deviation with respect to the mean of $-72.3955$ m, showing that it is a value that is well below the mean and, therefore, that the distances are greatly reduced. The value obtained for $\alpha$ represents an increase in the cost of 3 to 7 times for vertical versus horizontal distances, indicating that these have a higher cost and should therefore be reduced by a greater amount. In addition, being a high value within the scale, it minimizes the vertical distances traveled by a large amount, reaching one of its lowest values at 168 m. Finally, this is the optimum value selected for the calibration of $\alpha$ in the experiment.

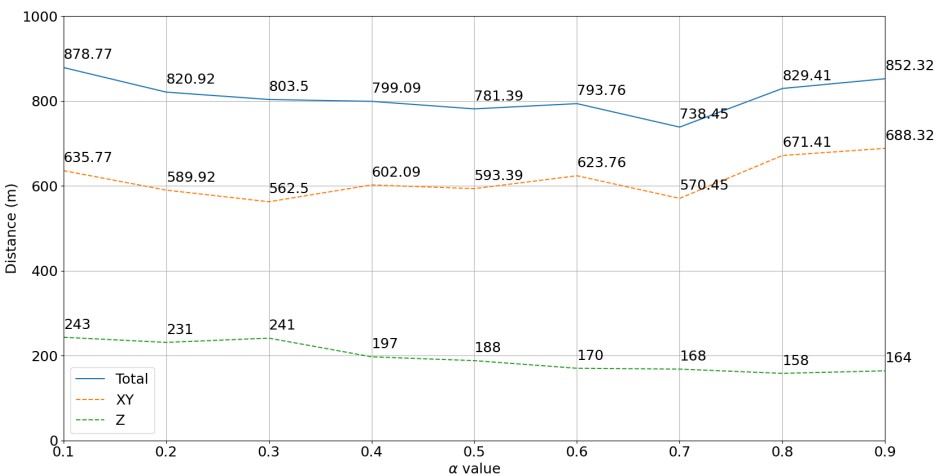

**Figure 3.** Total traveled distances at the variation of $\alpha$ penalty factor.

*6.3. Results*

Next, the reached results are presented by both methods in the scenarios described. In addition, the obtained routes are detailed for scenarios 8, 20, and 128 nodes. These quantities are selected to compare the model's effectiveness and the methods in networks of different sizes. It is important to remember that the workstations visited by the drones are random; it is for solely satisfying the same quantity of stations per floor.

6.3.1. Method Comparison and Results

The comparison of the methods with the values obtained in the tests is detailed in Table 4. This table presents the energy consumption cost (CE), the processing time (T), and the relative error (RE), i.e., the difference between the obtained energy cost by the optimal solution and the heuristic. The consumption costs are valued in dollars to use a common standard; these costs are only referential for the comparison.

$$\%RE = \frac{CE_{heuristic} - CE_{optimal}}{CE_{optimal}} \times 100\%$$

**Table 4.** Comparison between CPLEX with VRP *MinLe* model and heuristic.

| Scenario | n | m | CPLEX | | | Heuristic | | | |
|---|---|---|---|---|---|---|---|---|---|
| | | | M | CE (USD) | Time (s) | M | %RE | CE (USD) | Time (s) |
| 1 | 16 | 5 | 4 | 0.320 | 4.6 | 4 | 6.04 | 0.339 | 0.000 |
| 2 | 32 | 10 | 4 | 0.579 | 3600 | 4 | 25.49 | 0.726 | 0.001 |
| 3 | 48 | 10 | 5 | 0.845 | 3600 | 5 | 19.60 | 1.010 | 0.001 |
| 4 | 64 | 10 | 7 | 1.154 | 3600 | 7 | 20.04 | 1.386 | 0.001 |
| 5 | 80 | 10 | 10 | 1.541 | 3600 | 8 | 7.55 | 1.657 | 0.001 |
| 6 | 96 | 15 | 10 | 1.647 | 3600 | 7 | 15.26 | 1.898 | 0.001 |
| 7 | 112 | 15 | 10 | 1.99 | 3600 | 8 | 8.83 | 2.167 | 0.001 |
| 8 | 128 | 15 | 10 | 2.31 | 3600 | 9 | 11.97 | 2.597 | 0.001 |

Marked discrepancies can be observed between the performance of the exact solver and the heuristic in terms of execution times. Upon examination of Table 4, it becomes evident that CPLEX, a popular exact solver software, is found to reach the preset time limit of one hour in seven out of eight presented test cases. This is seen to contrast significantly with the agility of the heuristic, where superior temporal efficiency is displayed. Rather than a

full hour being dedicated to each problem, as is the case with CPLEX, the heuristic is found to address and resolve the same cases in just a fraction of the time, thus demonstrating its dramatically faster performance.

The effectiveness and precision of the results are considerably higher when the solver is used compared to the heuristic. Specifically, the results obtained through the CPLEX solver stand out for their noticeable superiority over the heuristic, even reaching a differential of up to 25.49% in favor of the solver in the most unfavorable scenario.

It is important to highlight that this gap is attributed to the suboptimal performance of the heuristic, which manifests itself in unsatisfactory results. This fact underscores the pressing need to develop and apply more sophisticated and efficient heuristics to deal with the problem at hand.

Finally, fewer drones are generally utilized by the heuristic as compared to the model. It was observed that in four instances, all available drones were used by CPLEX, while fewer were used by the heuristic. However, it is important to note that this does not necessarily imply a minimization of the total energy in the scenario. In fact, in scenario 6, even though 7 drones were employed, there was an RE of 15.26% in comparison with CPLEX. This indicates that the routes generated by the heuristic may necessitate longer distances to be covered, but this is offset by the usage of fewer drones.

An important aspect to highlight is the difference between energy costs obtained for each scenario. As expected, energy costs for optimal solutions are lower than those evaluated by the heuristic, with an average difference of around 13.035 %. On the other hand, the optimal solution takes, in most cases, nearly one hour to be obtained, while the heuristic solution takes only a few seconds.

6.3.2. Visual Analysis of Solutions

The assignment of routes to each drone varies between methods, but feasible routes for the problem at hand are generated by both. The solutions for scenario 6 are exhibited in Figures 4 and 5. Each route is assigned to a drone, with each color representing a floor and each number corresponding to a specific workstation. It is notable that the workstations are distributed almost equitably among the drones by the heuristic. This result is attributed to the KMeans algorithm, which generates balanced clusters. This same characteristic is observed in the scenario presented in Figures 6 and 7.

Finally, the results obtained for scenario 8 of 128 nodes are shown in Figures 6 and 7, which correspond to the instance with all the nodes of the studied problem.

In Figure 8, the nodes and floors (levels) visited by the respective drones from 1 to 15 are represented. From Figure 8, it can be observed that eight drones visit only one floor, while eight drones visit two floors. This graph validates the consistency of the model to assign drones to the minimum number of levels, as management desired, to facilitate a simple flight space design in the factory.

The heuristic implemented ensured an optimal allocation of drones across various floors. In Figure 9, the assignment of drones, represented in distinct colors, to each workstation on the different levels is illustrated. It is noted that in this specific context (Scenario 5), the first floor was serviced by two exclusive drones, while the maintenance of the rest of the floors required the intervention of more than two drones.

The ideal for the resulting routes is for the drone to leave the base, visit only one level to deliver the cargo, and then return to the warehouse. This is to have the minimum number of trips between levels. In Table 5, we present a summary of the number of levels visited by each drone, broken down by resolution technique, for Scenario 8.

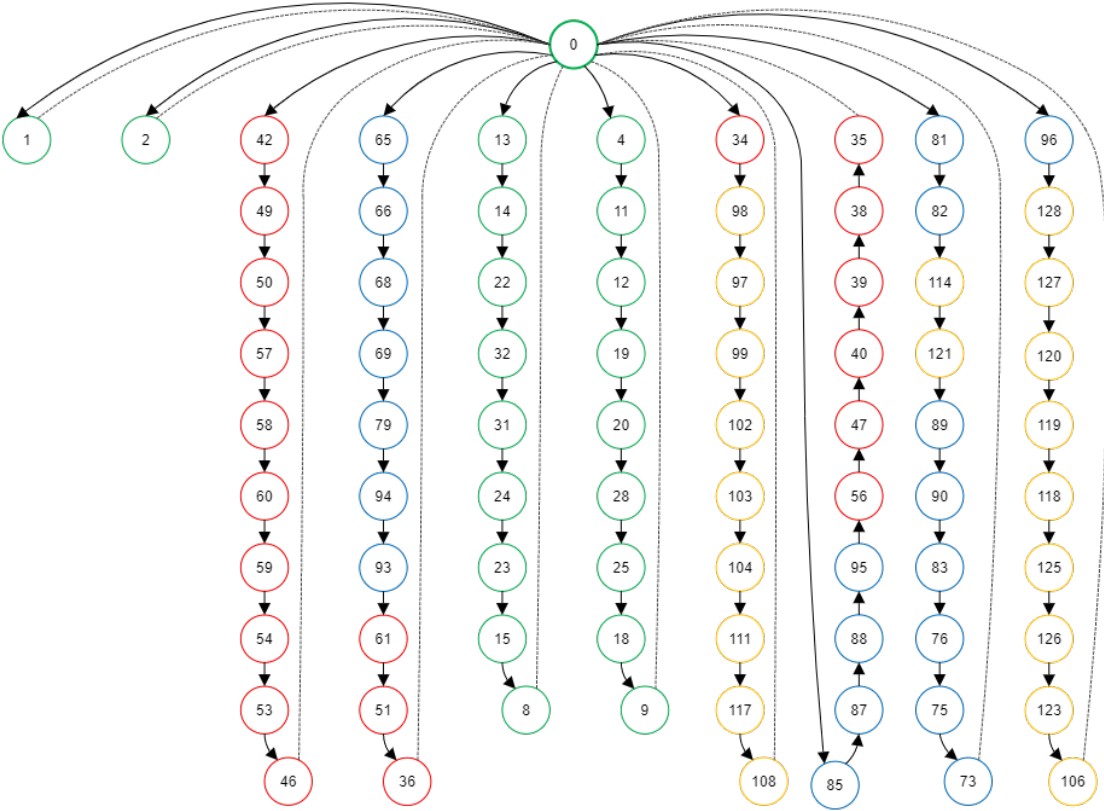

**Figure 4.** Optimal solution for scenario 6. The green color indicates the first floor, the red the second level, the blue color indicates the third level and the yellow color the fourth.

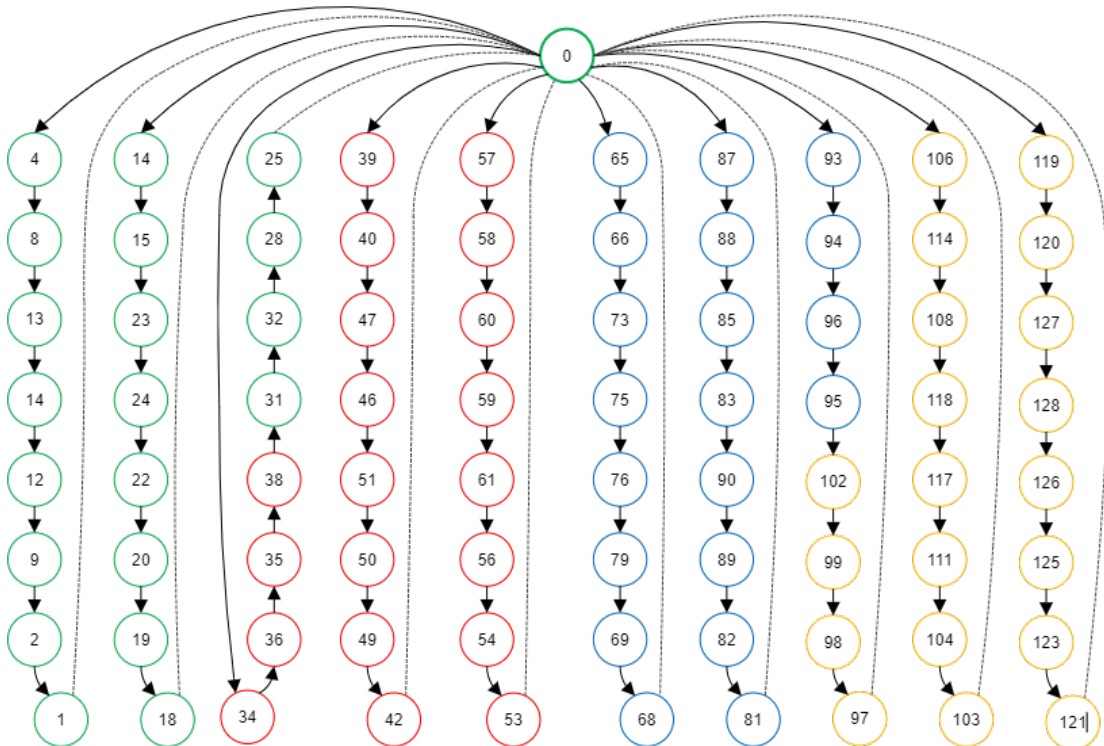

**Figure 5.** Heuristic solution for scenario 6. The green color indicates the first floor, the red the second level, the blue color indicates the third level and the yellow color the fourth.

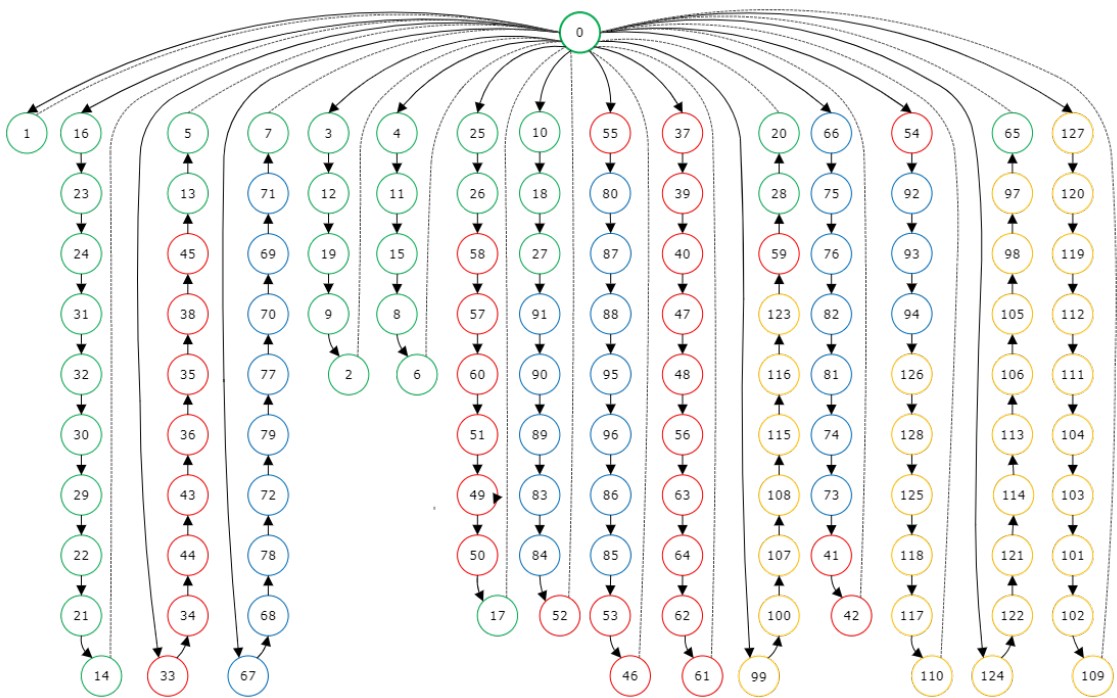

**Figure 6.** Optimal solution for scenario 9. The green color indicates the first floor, the red the second level, the blue color indicates the third level and the yellow color the fourth.

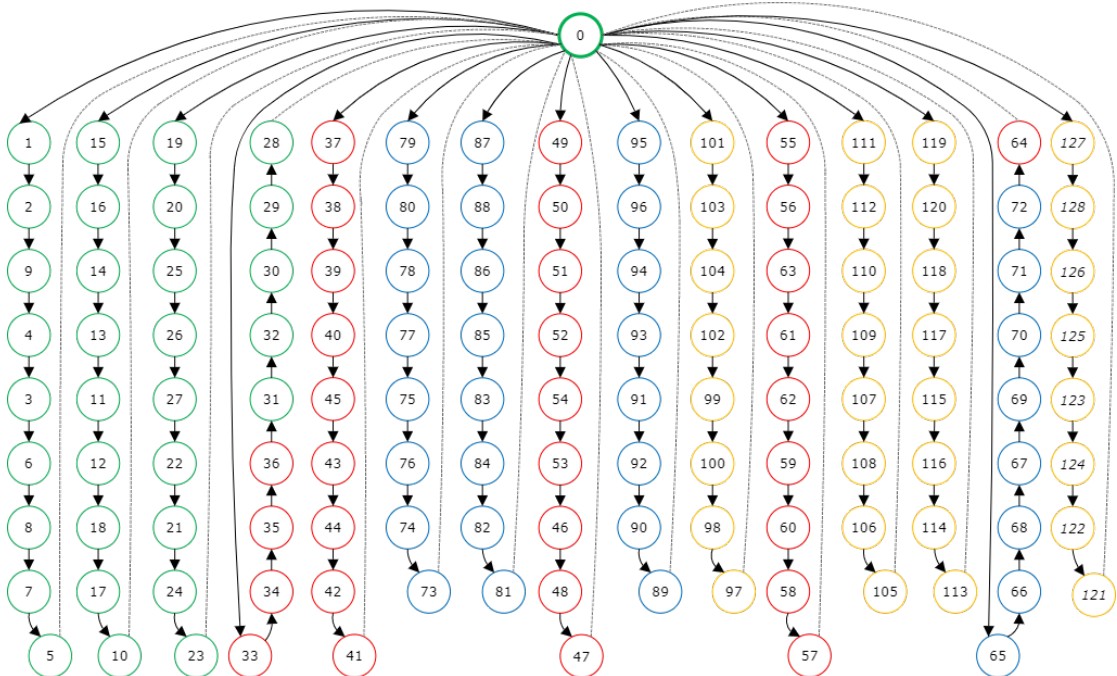

**Figure 7.** Heuristic solution for scenario 8. The green color indicates the first floor, the red the second level, the blue color indicates the third level and the yellow color the fourth.

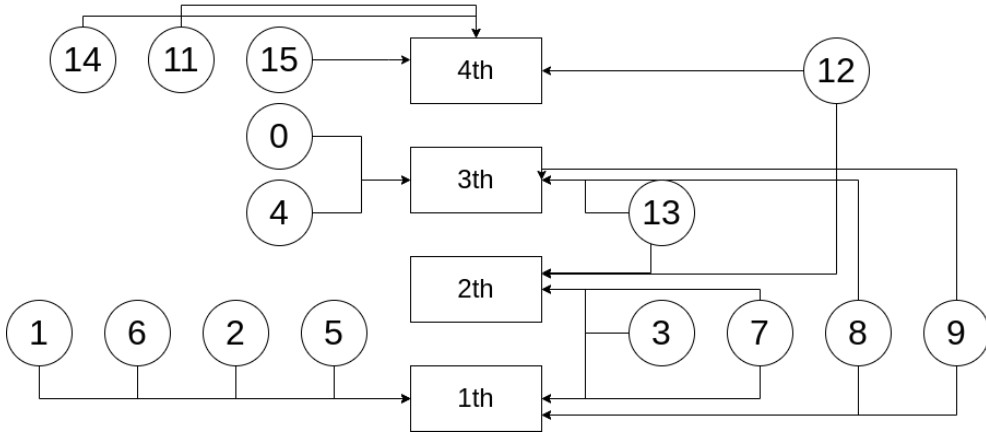

**Figure 8.** Floors (levels) visited by the respective drones.

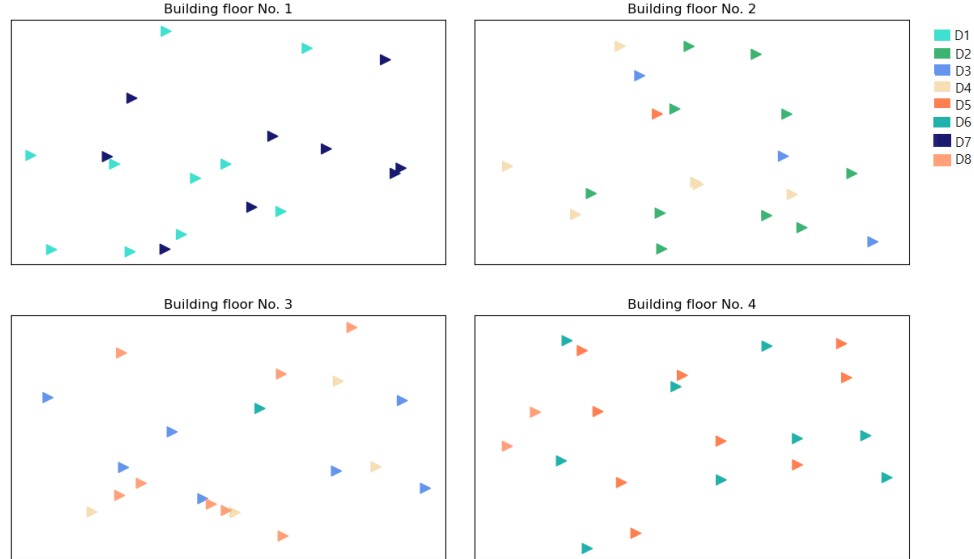

**Figure 9.** Distribution of drones per floor for scenario 5.

**Table 5.** Drone quantity according to the number of visited levels of every solution for scenario 8.

| Visited Levels | Optimal Solution | Heuristic Solution |
|---|---|---|
| One level | 4 | 12 |
| Two levels | 5 | 2 |
| Three levels | 6 | 0 |

## 7. Conclusions

This paper introduces a novel Vehicle Routing Problem (VRP) model, termed the Minimum Levels VRP. It builds on the 3D VRP model by [27], initially employed for material distribution tasks. The advanced formulation presents a new parameter designed to minimize the frequency of trips between factory levels for drones. This optimization trims down the total distance the drones cover and curtails the related energy costs. A crucial aspect of this model is the fine-tuning of the penalty factor, denoted as $\alpha$ for vertical distances. After a comprehensive analysis of the results, an optimal $\alpha$ value of 0.7 is determined by calculating optimal solutions. Pinpointing this optimal value is paramount to the model's successful execution as it significantly enhances the model's performance.

This paper employs the CPLEX software to solve the proposed model and secure optimal solutions. However, given the excessive and impractical resolution time observed,

the heuristic technique LKH emerges as a viable alternative. This heuristic yields feasible solutions, which are compared with the CPLEX results on critical implementation aspects, including variations in energy consumption, disparities between optimal and heuristic routes, and the number of floors the drones visit.

The routes derived from each method significantly differ. For more extensive networks, the heuristic method demonstrates superior efficacy, effectively minimizing the number of levels traversed by the drones. A detailed analysis of the results obtained for a 128-node scenario corresponding to the original factory problem reveals intriguing insights. The energy cost of the heuristic solution rises by 11.97% relative to the optimal solution. This increase is notably lower than the general average, leading to substantial energy savings and averting mid-air drone discharges.

One of the primary objectives of the problem—reducing the number of floors visited by the drones—is achieved remarkably well in this scenario, with 13 out of every 15 drones remaining on a single floor for distribution. Therefore, the heuristic emerges as the more efficient of the two methods for resolving this problem. Not only does it locate optimal solutions with high precision, but it also fulfills a key problem objective by decreasing inter-floor trips, resulting in an average energy consumption increase of only 13.5%.

Though this increase may initially seem significant, it becomes less relevant when considering the superior quality of the heuristic solution. This solution encapsulates various criteria, including a reduction in inter-level trips, energy conservation, and efficient route search execution. Moreover, it is practical and ready for implementation.

The complexity of routing problems requires certain simplifications in our model. Future work could account for in-flight battery or fuel limitations, especially in larger factories or logistic centers, in contrast to our medium-sized factory scenario. Drone reusability, another important factor, should have been considered in our problem, in which we assumed a fleet size sufficient for demand. Given the high cost of drone technology, studying scenarios with insufficient drones for concurrent demands could be valuable.

Furthermore, new resolution methods should be explored to improve results and accommodate high-demand processes. Although cluster formation was straightforward in this paper, the literature suggests more efficient node grouping and route assignment methods. By addressing these factors, we aim to refine our model and its detailed results for practical implementation, enhancing its usability across various industries and improving logistics.

**Author Contributions:** Conceptualization, I.D. and C.R.; methodology, I.D.; software, C.R.; validation, I.D. and C.R.; formal analysis, C.R.; investigation, I.D.; resources, I.D.; data curation, C.R.; writing—original draft preparation, I.D.; writing—review and editing, C.R.; visualization, C.R.; funding acquisition, I.D. All authors have read and agreed to the published version of the manuscript.

**Funding:** This research was funded by The Industrial Engineering Dept. USACH. The authors also gratefully acknowledge the support of the Faculty of Engineering of the University of Santiago, Chile, and the Center of Operations management and operations research CIGOMM. Furthermore, we extend our thanks to the Agencia Nacional de Investigación y Desarrollo (ANID), whose financial support through the "Subvención a la Instalación en la Academia" program, file 85220108, has been instrumental in the realization of this project. Finally, we are grateful to the anonymous reviewers for their valuable comments that helped improve the quality of our work.

**Data Availability Statement:** Not applicable.

**Conflicts of Interest:** The authors declare no conflict of interest.

## Appendix A

**Table A1.** Obtained results for scenario 1, 16 nodes.

| Drone | Optimal Solution | Heusristic Solution (LKH) |
|---|---|---|
| Drone 1 | 0-1-0 | 0-34-35-40-0 |
| Drone 2 | 0-35-75-66-34-0 | 0-66-75-56-0 |
| Drone 3 | 0-12-19-25-00-1-0 | 0-1-25-19-12-0 |
| Drone 4 | 0-83-96-56-40-0 | 0-97-83-96-0 |
| Drone 5 | 0-97-125-103-0 | 0-103-118-125-0 |

**Table A2.** Obtained results for scenario 2, 32 nodes.

| Drone | Optimal Solution | Heusristic Solution (LKH) |
|---|---|---|
| Drone 1 | 0-1-0 | 0-14-12-9-1-0 |
| Drone 2 | 0-9-0 | 0-3935-34-0 |
| Drone 3 | 0-19-12-0 | 0-40-47-50-0 |
| Drone 4 | 0-14-56-31-22-0 | 0-66-58-56-0 |
| Drone 5 | 0-25-58-50-34-0 | 0-73-81-75-0 |
| Drone 6 | 0-3547-40-29-0 | 0-19-22-31-25-0 |
| Drone 7 | 0-75-81-73-66-0 | 0-88-93-83-0 |
| Drone 8 | 0-88-96-93-83-0 | 0-96-102-97-0 |
| Drone 9 | 0-97-102-111-103-0 | 0-118-111-103-0 |
| Drone 10 | 0-118-128-125-121-0 | 0-121-125-128-0 |

**Table A3.** Obtained results for scenario 3, 48 nodes.

| Drone | Optimal Solution | Heusristic Solution (LKH) |
|---|---|---|
| Drone 1 | 0-19-83-81-73-9-0 | 0-4-14-12-9-1-0 |
| Drone 2 | 0-14-22-32-31-24-0 | 0-18-25-19-22-24-0 |
| Drone 3 | 0-25-58-59-50-18-0 | 0-34-35-36-32-0 |
| Drone 4 | 0-1-12-4-0 | 0-39-40-47-46-50-0 |
| Drone 5 | 0-36-68-75-66-34-0 | 0-66-58-59-56-53-0 |
| Drone 6 | 0-35-46-47-40-39-0 | 0-68-75-76-81-73-0 |
| Drone 7 | 0-53-118-108-98-97-0 | 0-111-119-118-108-0 |
| Drone 8 | 0-76-93-94-96-88-0 | 0-121-126-128-125-0 |
| Drone 9 | 0-56-128-126-125-121-0 | 0-87-88-94-93-83-0 |
| Drone 10 | 0-102-103-111-119-87-0 | 0-96-103-102-98-97-0 |

**Table A4.** Obtained results for scenario 4, 64 nodes.

| Drone | Optimal Solution | Heusristic Solution (LKH) |
|---|---|---|
| Drone 1 | 0-1-0 | 0-4-8-13-14-12-9-0 |
| Drone 2 | 0-8-40-39-47-46-36-4-0 | 0-18-25-28-24-22-20-19-0 |
| Drone 3 | 0-9-34-42-50-57-25-18-0 | 0-65-66-57-58-59-61-0 |
| Drone 4 | 0-22-24-31-32-28-20-13-0 | 0-68-75-76-82-81-73-0 |
| Drone 5 | 0-12-108-111-104-103-102-35-0 | 0-34-36-3539-40-31-32-0 |
| Drone 6 | 0-19-51-59-61-56-53-14-0 | 0-87-88-93-94-89-83-0 |
| Drone 7 | 0-66-65-97-98-114-121-58-0 | 0-96-95-103-102-98-97-0 |
| Drone 8 | 0-73-81-82-89-94-93-83-0 | 0-87-88-95-96-76-75-68-0 |
| Drone 9 | 0-87-88-95-96-76-75-68-0 | 0-108-114-118-119-111-104-0 |
| Drone 10 | 0-118-125-126-128-127-120-119-0 | 0-120-127-128-126-125-121-0 |

**Table A5.** Obtained results for scenario 6, 96 nodes.

| Drone | Optimal Solution | Heusristic Solution (LKH) |
|---|---|---|
| Drone 1 | 0-1-0 | 0-1-2-9-11-4-8-7-0 |
| Drone 2 | 0-2-0 | 0-18-19-12-14-16-15-13-0 |
| Drone 3 | 0-4-0 | 0-23-24-21-22-20-26-25-0 |
| Drone 4 | 0-9-0 | 0- 56-61-59-60-58-57-0 |
| Drone 5 | 0-11-0 | 0-69-70-68-73-66-65-0 |
| Drone 6 | 0-8-15-16-23-24-31-32-22-21-14-0 | 0-33-34-35-36-31-32-28-0 |
| Drone 7 | 0-33-65-97-99-107-108-113-106-98-34-0 | 0-81-82-76-75-80-79-0 |
| Drone 8 | 0-13-46-53-47-40-39-7-0 | 0-89-83-85-86-88-87-0 |
| Drone 9 | 0-35-36-68-69-70-79-80-87-45-37-0 | 0-37-38-39-40-45-46-42-0 |
| Drone 10 | 0-12-20-28-26-25-0 | 0-95-96-94-93-91-90-0 |
| Drone 11 | 0-42-49-50-57-58-60-59-52-51-19-0 | 0-101-103-102-99-98-97-0 |
| Drone 12 | 0-18-82-89-90-91-83-81-73-66-0 | 0-49-50-51-52-54-53-47-0 |
| Drone 13 | 0-56-88-120-119-111-104-103-101-102-38-0 | 0-106-113-108-107-111-104-0 |
| Drone 14 | 0-54-61-93-94-96-95-86-85-76-75-0 | 0-119-120-117-118-121-114-0 |
| Drone 15 | 0-127-128-126-125-118-117-123-124-121-114-0 | 0-123-124-125-126-128-127-0 |

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
