# Peer review of "Drone Optimization in Factory: Exploring the Minimal Level Vehicle Routing Problem for Efficient Material Distribution"

_drones, doi:10.3390/drones7070435_

Round 1
Reviewer 1 Report
This paper presents a new model for minimal horizontal vehicle routing planning, which is primarily designed for the distribution of materials using drones. The main aim of the proposed model is to minimize the travel distance covered by drones while also reducing the number of factory floors visited during each delivery from the base. Additionally, the author introduces a penalty factor for vertical movement to decrease the vertical travel distance and enhance the overall efficiency of the model. The paper explores both exact and heuristic techniques to solve the problem, compare their performance, and conclude that the heuristic technique provides a faster solution while still achieving the desired reduction in vertical traveling distances. Finally, the author concludes that this newly proposed minimal level VRP model has immense potential in improving the efficiency of drone-based material distribution within factories and thereby reducing operational costs.
The authors' study is interesting, and their methods are applicable. However, the paper's writing could be improved and revised. The detailed comments are as follows:
1、Abstract should be clear and check whether all the points mentioned in abstract are addressed in this manuscript.
2、To highlight the novelty of this paper, the comparison with existing studies needs to be further strengthened, such as: Sub-super-stochastic matrix with applications to bipartite tracking control over signed networks,Opinion Polarization Over Signed Social Networks With Quasi Structural Balance, A Cooperation-Competition Evolutionary Dynamic Model Over Signed Networks .
2、The manuscript's experimental results are somewhat less intuitive. The author could add some line graphs or add a few different examples to reflect the advantages and disadvantages of different methods in different indicators. If possible, the advantage of proposed algorithm could be represented by the percentage improvement of the indicator;
3、Some formula symbols in the article lack explanations. It is recommended that authors could read through the full manuscript to make some explanations about the first occurrence of the variable in the formula.
4、The layout of images and formulas in the paper could be further optimized for better understanding by the reader.
No comment
Author Response
Drone Optimization in Factory: Exploring the Minimal Level Vehicle Routing Problem for Efficient Material Distribution
The following responses to the concerns of the reviewers of the scientific article are presented below:
Reviewer 1
- Abstract should be clear and check whether all the points mentioned in abstract are addressed in this manuscript.
A: Thanks for your observation. The abstract was totally restructured according to the study paper.
- To highlight the novelty of this paper, the comparison with existing studies needs to be further strengthened, such as: Sub-super-stochastic matrix with applications to bipartite tracking control over signed networks,Opinion Polarization Over Signed Social Networks With Quasi Structural Balance, A Cooperation-Competition Evolutionary Dynamic Model Over Signed Networks .
A: Thanks for the observation. A subsection was developed in the introduction called "Drones inside the factory in the literature". At this point, a table was created with the main papers found with the phrase drones in factory.
- The manuscript's experimental results are somewhat less intuitive. The author could add some line graphs or add a few different examples to reflect the advantages and disadvantages of different methods in different indicators. If possible, the advantage of proposed algorithm could be represented by the percentage improvement of the indicator.
A: Thanks for your advice. Two new figures have been added to illustrate the effective effect of the heuristics.
- Some formula symbols in the article lack explanations. It is recommended that authors could read through the full manuscript to make some explanations about the first occurrence of the variable in the formula.
A: Thanks for your observation. Both the explanation of variables and the explanation of the model were completely rewritten.
- The layout of images and formulas in the paper could be further optimized for better understanding by the reader.
A: Thanks for your advice. Both the formulas and images were improved in this version.

Reviewer 2 Report
1. I’d suggest spending a couple more lines on CRP and ML-VRP in abstract, in order to increase its “appeal”.
2. I think figures 2-4 are almost useless, maybe merging them with #1 is better.
3. Lines 182-191 could probably be more effective if interleaved with their respective constraints.
4. Objective function before eq. 17 should be numbered and it’s badly formatted.
5. Why equal distribution of workstation among floors has been chosen?
6. Is alpha = 0.7 a general result or does it depend on the specific problem instance?
7. I think tables are mildly redundant.
8. Overall, the main weakness of this work is the originality in comparison to 3D-VRP which must be substantiated as strongly as possible.
I find this work quite easy to read.
Author Response
Drone Optimization in Factory: Exploring the Minimal Level Vehicle Routing Problem for Efficient Material Distribution
The following responses to the concerns of the reviewers of the scientific article are presented below:
Reviewer 2
- I’d suggest spending a couple more lines on CRP and ML-VRP in abstract, in order to increase its “appeal”.
A: Thanks for your observation. The abstract was totally restructured according to the study paper.
- I think figures 2-4 are almost useless, maybe merging them with #1 is better.
A: Thank you very much for your advice. Figures 2 and 3 have been removed.
- Lines 182-191 could probably be more effective if interleaved with their respective constraints.
A: Thanks for your observation. The explanations for each constraint have been placed interspersed.
- Objective function before eq. 17 should be numbered and it’s badly formatted.
A: Thank you so much. We have corrected this point.
- Why equal distribution of workstation among floors has been chosen?
A: Since the floors are the same with the same surface area, 32 jobs were assigned per floor, but they are grouped by processes within the manufacturing process, following the logic of the product flow. Thus, on the fourth floor the leather is cut, on the third floor the upper part of the shoe is sewn, on the second the sole and heel of the shoe are made, while on the first floor the shoe is assembled by joining the upper and sole and heels.
- Is alpha = 0.7 a general result or does it depend on the specific problem instance?
A: The value a=0.7 was obtained from a study in which he solved the optimal assignment problem for different values from 0.1 to 0.9 by varying 0.1 at a time. The minimum cost was obtained for a=0.7, therefore it must be considered a result.
- I think tables are mildly redundant.
A: Thanks for the observation. The tables were totally removed.
- Overall, the main weakness of this work is the originality in comparison to 3D-VRP which must be substantiated as strongly as possible.
A: Added a subsection within the introduction explaining the differences with the previous article. Call Our contribution

Reviewer 3 Report
Comments are included in the attached document

Author Response
Drone Optimization in Factory: Exploring the Minimal Level Vehicle Routing Problem for Efficient Material Distribution
The following responses to the concerns of the reviewers of the scientific article are presented below:
Reviewer 3
- Overall comment: What people do in a given time, how long would it take with drones? Does it make sense to use drones for that? Once this question has been answered, then you can begin to formulate the model to calculate the optimal trajectories.
A: This excessive time is due to two effects, first the dealers attend to problems reported by operators at the beginning of the shift such as lack of tools, minor failures of machines, lack of materials, etc. and second, time is lost in conversations personal.
- In your factory, currently 8 people do it, 2 per floor and it takes 2 hours to deliver and 2 hours to collect. How much is saved with drones?
A: Thank you. The following paragraph was added in line 189: Considering that a worker of this type in Chile represnts a company expense of 9.000 euros annually. It can be established that the saving of 8 workers due to the use of drones brings a saving of 72.000 euros per year.
- it is said that “fourth floors have identical workstation layouts”. Perhaps saying that and that the numbering continues from one floor to the next one is enough.
A: Thank you very much for your advice. Figures 2 and 3 have been removed.
- Line 198. “These costs are modelled by equations (1) and (2)”. Equations numbering (1) & (2) are right?
A: Thank you. This point was corrected.
- Line 198. Typo: modelled instead modeled
A: Thank you, the sentence was restructured.
- Line 200 and next: ….to simplify (1) and (2), we obtain (3) and (4)”. Is this numbering right?
A: Thank you, the sentence was restructured.
- Line 201: Vij is the horizontal velocity of the drone in arc ij and in Line 181: Vij : distance from node i to node j in the vertical axis. Use same term for different parameter does not help to understand the text.
A: Thank you, the formulations were completely restructured.
- After equation (22). In addition, the constraint (123). Please clarify what is constraint (123)
A: Thank you, the formulations were completely restructured.
- Line 229. Please provide the meaning of the acronyms the first time they apperar along the text.
A: Thank you, the sentence was restructured.
- LKH (Lin-Kernighan Heuristic)……….. to address the travelling sales person (TSP
A: Thank you, the sentence was restructured.
- Line 266. Always “ is used along th text. However, at this time, “alpha” is used
A: Thank you, the sentence with “alpha” was restructured.
- Table 1: Indicate units: meters?
A: Thanks for the observation. Units of measure have been added to the new table.
- Line 294: “First, the visited quantity of nodes of a drone in the optimal solution of the VRP is observed to be equal to the ones visited in the heuristic”. Observing table 6 is not true for scenarios 1, 6,7,8 & 9.
A: Thanks for your observation. The paragraph has been removed as the sentence is false.
- Line 297. Typo: basing it on instead basing it in.
A: Thank you, the sentence was restructured.
- Line 299: “As expected, energy costs for optimal solutions are lower than those evaluated by the heuristic,….” Not for scenario 1. Why is it used absolute values in RE% equation?. Negative sign means that heuristic CE is lower than optimal CE.
A: Thanks for your observation. Absolute value has been removed from the equation.
- Table 6: Computation time is the same for 32 nodes with 10 drones and 128 nodes with 15 drones, is that right?
A: Exactly. The solver timeout value has been set to 1 hour. We have added a sentence in the experimental protocols.
- Table 6: CE (USD) column: what does this figure mean? Is the total energy cost for all scenario?: I mean the energy consumed by all drones which work in a particular scenario when they are running.
- A: Exactly. But Table 6 is now Table 4.
- Line 331: Typo: …..emerges as a feasible alternative.
A: Thank you, the sentence was restructured.

Round 2
Reviewer 1 Report
N/A
N/A
Author Response
Drone Optimization in Factory: Exploring the Minimal Level Vehicle Routing Problem for Efficient Material Distribution
The following responses to the concerns of the reviewers of the scientific article are presented below:
Reviewer 1
|
|
Must be improved |
|
|
|
Does the introduction provide sufficient background and include all relevant references? |
(x) |
|
( ) |
Answer:
We appreciate the observation "must be improved" regarding the introduction. We have enriched the text by adding two paragraphs to meet the requirements. The first one explores the future potentials of drones in manufacturing processes based on the advantages these devices offer today. The second new paragraph introduces one of the most significant novelties of our study, describing the impacts of including a new index in this type of mathematical problem.
We hope these adjustments meet the requirements and provide a complete understanding of our work.

Reviewer 2 Report
Thank you for solving most of the points I raised. I've just two main concerns:
*novelty - I'm not sure the contribution is enough for a paper, anyway the choice is up to the editor
*if alpha = 0.7 is a general result you should provide a sensitivity analisys by changing the problem details and proving that alpha remains 0.7, otherwise you could simply present the test you already did without presenting it as a general result
Although English can surely be improved I find quite easy to read this work.
Author Response
Drone Optimization in Factory: Exploring the Minimal Level Vehicle Routing Problem for Efficient Material Distribution
The following responses to the concerns of the reviewers of the scientific article are presented below:
Second round of review
Reviewer 2
- *novelty - I'm not sure the contribution is enough for a paper, anyway the choice is up to the editor
We appreciate your comment highlighting the need to emphasize the novelty of our article. We have incorporated two new paragraphs into the manuscript to accentuate our contribution. The first, located in the introduction (lines 75 -80), provides an overview of the findings when introducing modifications to the mathematical model. The second paragraph (lines 167-173) clearly articulates our model's effective impact.
It should be noted that, although these paragraphs do not delve into the experimental part, the results section provides a quantitative evaluation of the model's performance and showcases some route layouts in the context of the studied manufacturing plant (for example, see Figure 8).
We trust that the additions meet your requirements and enhance the relevance and originality of our research.
- If alpha = 0.7 is a general result you should provide a sensitivity analysis by changing the problem details and proving that alpha remains 0.7, otherwise you could simply present the test you already did without presenting it as a general result.
Thank you for your observation. This suggestion is very interesting, and we would like to include this analysis, but it is impossible in the 5 day period that we have been given.
In addition, we have conducted specific experiments to determine the best value of the alpha parameter. The results of these experiments are illustrated in Figure 3 and are discussed in depth in lines 338-348. Although we do not provide an exhaustive analysis, we present sufficient statistical data to justify our choice of the alpha parameter.
We hope this additional information will be well received and contribute to a greater understanding of our approach and methodology
